# Investigation of Biofilms Formed on Steelmaking Slags in Marine Environments for Water Depuration

**DOI:** 10.3390/ijms21186945

**Published:** 2020-09-22

**Authors:** Akiko Ogawa, Reiji Tanaka, Nobumitsu Hirai, Tatsuki Ochiai, Ruu Ohashi, Karin Fujimoto, Yuka Akatsuka, Masanori Suzuki

**Affiliations:** 1National Institute of Technology (KOSEN), Suzuka College, Shiroko-cho, Suzuka, Mie 510-0294, Japan; hirai@chem.suzuka-ct.ac.jp (N.H.); ochiaitatsuki@gmail.com (T.O.); ruusama0624@yahoo.co.jp (R.O.); hujimoto.masakazu@snow.plala.or.jp (K.F.); bumpzero@outlook.jp (Y.A.); 2Graduate School of Bioresources, Mie University, 1577 Kurimamachiya-cho, Tsu, Mie 514-8507, Japan; tanakar@bio.mie-u.ac.jp; 3Graduate School of Engineering, Osaka University, 1-1 Yamadaoka, Suita, Osaka 565-0871, Japan; suzuki@mat.eng.osaka-u.ac.jp

**Keywords:** steelmaking slag, biofilm, microbiome, bioremediation, iron and steel slag

## Abstract

Steelmaking slags are a promising resource as artificial seaweed beds for the reconstitution of marine environments. To grow seaweed well, the formation of biofilms is an essential process in biofouling. This study focused on the formation of initial biofilms on steelmaking slag samples and analyzed the resulting bacterial communities using the next-generation sequencing technique. Three types of steelmaking slag were submerged in an area of Ise Bay in Mie Prefecture, Japan, for 3 and 7 days in the summer and winter seasons to allow the formation of biofilms. The bacterial communities of these biofilms were richer in sulfur-oxidizing bacteria compared to the biofilms formed on polyurethane sponges. It was found that Helicobacteraceae dominantly grew on the biofilms formed on the slag samples. This shows that steelmaking slags have potential to be used as artificial seaweed beds and marine water purifiers.

## 1. Introduction

Almost all iron and steel slags are used as a raw material for cement production, concrete aggregates, ground improvements, and so on [1]. Japan produced 10,930 kt of steelmaking slag in the 2017 fiscal year, which was mainly used as raw material in civil engineering applications (31%) and for road constructions (32%) [2]. However, in these applications, there are other alternative resources of such raw materials; therefore, there is a need for novel applications of these by-products. Steelmaking slags contain mainly lime (CaO), silica (SiO_2_), metallic oxidized iron (t. Fe), and magnesia (MgO) [3], along with smaller quantities of sulfur (S), oxidized manganese (MnO), oxidized phosphorus (P_2_O_5_), and alumina (Al_2_O_3_). Fe, S, and SiO_2_ are essential elements for phytoplankton, however, Fe tends to be scarce in marine environments [4]. Therefore, steelmaking slag is a potential source of phytoplanktonic nutrients. Numata et al. reported that diatoms and green algae were successfully grown on steelmaking slag immerged in a live marine environment for four years [5]. Arita et al. cultured marine phytoplankton in a marine broth containing microparticles of steelmaking slag, which showed that steelmaking slag is a potential source of iron, phosphorus, and silicon [4]. In JFE steel Co., Miyata et al. performed long-term verification tests on steelmaking slag employed in a marine block in In-no-Shima Island (Hiroshima, Japan) for several years. They found that the steelmaking slag was not only a suitable bed (basement) for growing algae but also an effective source of nutrients for them [6,7,8]. Additionally, they observed various benthic organisms, including rare species and fishes, in the tested area [9], which suggests that steelmaking slag is a good candidate for marine blocks. Additionally, Kamei et al. reported that a carbonated solid having an aragonite-type structure made from steelmaking slag [10] has good properties (comparable to those of concrete blocks) for seaweed or fish beds [11]. Yamamoto et al. focused on the eluted Fe (ions) from steelmaking slag to make up for dissolved Fe in the ocean. They combined steelmaking slag with humic substrates to prolong the iron elution, which successfully enhanced the growth of algae at over 20 coasts in Japan [12,13,14]. With these, steelmaking slag is thought to have a very good prospect for artificial seaweed beds. However, steelmaking slags cause white turbidity just after being immersed in the sea because of their free lime content. Free lime dissolves in seawater, which causes an increase in pH, thereby resulting in the conversion of the magnesium ions in the seawater into magnesium hydroxides, which are responsible for the turbidity in the seawater. Therefore, controlling the initial elution behavior of steelmaking slags is vital in marine block applications.

Biofouling is a vital process in the (re)generation of seaweed beds in marine environments. It involves the growth and attachment of organisms from bacteria to macroalgae and shellfish [15]. The first stage of biofouling is the immediate development (within a few seconds) of a conditioning film consisting of proteins and carbohydrates [16]. Next, floating planktons are irreversibly attached. They proliferate and secrete extracellular polymeric substrates (EPS) referred to as biofilms. In a marine environment, biofilms take about 1 month to mature. The biofilms enable the attachment of macroorganisms such as young mussels, barnacles, and macroalgae [17]. Therefore, the formation of biofilms is a very important step in the establishment of seaweed beds. In addition, it is considered that the early stage of biofilm formation strongly affects the white turbidity of steelmaking slag.

In this study, we investigated the relatively early stage (several days) of biofilm formation in three types of steelmaking slag submerged in a live marine environment. First, the slag samples were immersed in an area of Ise Bay in Japan, where biofilm formation was observed by Raman spectroscopy. Then, next-generation sequencing (NGS) technology was employed to analyze the formed biofilms, where both the culturable and unculturable bacteria could be detected. 

## 2. Results and Discussion

### 2.1. Megascopic Observation of Immerged Samples

In marine environments, the formation of biofilms on a steel substrate takes one month [18]. The early stage of biofilm formation, i.e., bacterial colonization, progresses over 24 h to one week. Herein, three different practical steelmaking slag samples were employed in the marine immersing test. The slags were in limited supply; therefore, we could not perform a pretest to determine the precise immersing periods. In this study, we focused on the early stage of biofilm formation on the surface of the steelmaking slag in the ocean. At the same time, we needed to obtain a sufficient amount of DNA from a single sample to analyze the bacterial flora formed in the biofilm; although the early stage of biofilm would contain a small amount of bacterial DNA, we could fail to isolate it. Additionally, we were challenged to investigate the change in bacterial flora. Considering these experimental conditions, we chose two immerging periods—three days and seven days. Figure 1 shows the appearance of the samples immerged during summer. As shown in the figure, there was a change in color in the polyurethane sponges (from light yellow to green–brown), where slime-like dirt was observed. It is also observed that the color-changed area of the seventh-day samples was larger than that of the third-day samples. The slime-like dirt was presumed to be a biofilm. The color of slag I samples also changed from dark gray to black. Their surfaces were covered with slime-like dirt, and their corners looked similar to wany. The seventh-day slag I samples looked more slippery than the third-day samples. Slag II samples also changed from dark gray to black and partly red–brown. The red–brown parts were supposedly red rusts, and they were more pronounced in the seventh-day slag II samples than the third-day ones. Slag II samples were partly covered with slime-like dirt. Slag III samples changed from light gray to dark gray and partly black. The black areas looked similar to slippery dirt, and they were supposedly biofilms.

Figure 2 shows the appearance of the samples immerged during winter. They looked very similar to those immerged in summer. However, the winter-immerged samples showed less amount of the slime-like dirt than the summer-immerged samples. Thus, it is considered that the formation of biofilms in the test area is slower in winter than in summer.

### 2.2. Identification of Biofilms by Raman Spectroscopy

Biofilms can be detected by crystal violet staining. Crystal violet dye is bound to negatively charged compounds of EPS components such as polysaccharides and nucleic acids [19,20]. However, this dye tends to be nonspecifically adsorbed to the pockets and indentations on the surfaces of materials, such as slags and sponges. Other methods of detecting biofilms include scanning electron microscopy (SEM), atomic force microscopy (AFM), and Raman spectroscopy [21]. In this study, slag I, II, and III were practical steelmaking slags without polishing, and they were immerged in the sea as received. The surfaces of these steelmaking slags were markedly indented to enable AFM observation. The SEM observation could not differentiate the biofilms from the surface of the steelmaking slags (data not shown). On the other hand, our research group has succeeded in detecting biofilms formed on an uneven surface such as corroded steels and polymer coating via Raman spectroscopy [22,23]. Therefore, we adopted Raman spectroscopy for biofilm detection. This technique can detect polysaccharides, proteins, lipids, and nucleotides, which are the main components of EPS derived from biofilms. Some researchers have employed Raman spectroscopy in the identification of biofilms [24,25,26,27,28,29,30,31]. 

Figure 3a–c present the Raman spectra of slag I, slag II, and slag III, respectively, after the marine immersion for 3 and 7 days in summer. As shown in Figure 3a,b for the 3 and 7 days, and 3c for the 7 days, two specific peaks are observed at around 1510 and 1150 cm^–1^. The peak at around 1510 cm^−1^ is attributed to a C=O stretch vibration of peptide linkages [32], and that at 1150 cm^−1^ corresponds to C–O–O stretching vibrations of lipid [33] and fatty acid stretching vibration of C–C [33]. These peaks are very similar to those of the normal resident biofilms (environmental bacterial biofilms) formed on glass- and silane resin-coated specimens in a laboratory biofilm reactor [29,34]. Therefore, these peaks are attributed to biofilm debris on the biofilms. Other remarkable peaks are observed at approximately 970 cm^−1^, which correspond to the P–O stretch of nucleic acid [32] in slag I (Figure 3a) and at around 1080 cm^–1^, corresponding to the C–O–O stretching vibration of lipid [32] in slag III (Figure 3c). These are related to the slag samples since the as-received slag (before the immersion test samples) also showed these peaks.

Unfortunately, Raman spectroscopy failed to detect any specific Raman peaks corresponding to the biofilms in the winter-immerged samples. This could be attributed to the fact that the amount of biofilm formed on these samples were very small (below the detectable level).

### 2.3. Bacterial Biome Analysis of the Formed Biofilms on the Surface of Each Specimen

Figure 4 shows the compositions of family levels of the bacterial biomes observed on the surface of each of the samples immersed during the summer season. Minor families were combined in other bacterial communities. However, in slag I on day three, the major bacterial groups were not determined, so they are not shown in Figure 4. Nevertheless, in slag I on day three, the amounts of biofilms were similar to that of the other samples, including that of slag types II and III. A very weak DNA concentration (data are not shown) was observed when the DNA extraction of each slag sample was performed. We infer that there was insufficient DNA in the biofilms of slag I on day three.

Eight main families were observed in each sample: Alteromonadaceae, Desulfobacteraceae, Desulfobulbaceae, Flavobacteriaceae, Helicobacteraceae, Piscirickettsiaceae, Rhodobacteraceae, and Saprospiraceae. It can be observed that the bacterial flora of the sponge samples (control) on days three and seven were significantly different from that of the slag samples. Even in the same porous material, bacterial flora varied. The reason for the remarkable difference in the bacterial flora is still unclear, however, each bacterial flora could be characterized by the variations in the slag content.

A detailed comparison of the bacterial florae of the control and slag samples revealed that Helicobacteraceae was the most prevalent family in the slag samples. Nakagawa et al. reported that Helicobacteraceae is involved in the oxidation of sulfides in the ocean, and it is likely to contribute to the detoxification of sulfide compounds [35]. In addition, Grote et al. reported that Epsilonproteobacteria, including Helicobacteraceae, exhibit denitrification activity [36]. Carlström et al. reported that *Sulfurimonas*, a newly discovered genus of Helicobacteraceae, can oxidize sulfides [37]. Furthermore, Takai et al. reported that *Sulfurimonas* autotrophically oxidizes sulfides in their experiment using sulfide, sulfur, thiosulfate, and hydrogen as electron donors and nitrite, nitrate, and oxygen molecules as electron acceptors [38]. Additionally, *Sulfurimonas paralvinellae* can perform denitrification in an anaerobic or microaerobic environment using carbon dioxide as a carbon source [38]. 

Desulfobulbaceae was observed as the second typical family in the slag samples. The members of this group were identified as organic acid oxidizers, which help in bacterial water purification. When in contact with water, they transport electrons from hydrogen sulfide-rich sediments to oxygen-rich sediments [39]. Desulfobulbaceae forms cable-like structures, which have been reported in a wide variety of sediments [40]. These structures may be useful as substrates in the first stage of biofilm formation.

Piscirickettsiaceae was also discovered as one of the typical families in the slag samples. These bacteria have been intensively studied for various applications, including water treatment, bioleaching, and bioremediation. Recent investigations revealed that these bacteria are S-oxidizing compounds [41].

Interestingly, all the samples, including the control samples, contained Saprospiraceae, under which 14 known bacteria have been classified [42]. The members of this family are essential in the breakdown of complex organic compounds in marine environments. In particular, *Saprospira* has both a high sulfide-oxidizing activity and an ability to degrade sulfate compounds in shallow-water gas vents [43].

Table 1 summarizes the compositions of the bacteria in the genera level for Helicobacteraceae, Desulfobulbaceae, and Saprospiraceae. Among them, approximately 30–50 bacteria were categorized as unknown. Therefore, they were classified under one group termed “unassigned” genera. For the Helicobacteraceae family, *Sulfurimonas* was the only identified main genus in all the three slag samples, however, it was not observed in the sponge samples. For the Desulfobulbaceae family, all bacteria were unassigned in all the samples. On the other hand, *Lewinella* was the main genus in Saprospiraceae in all the samples, and the dominance was higher in all the slag samples than in the sponge samples. *Lewinella* sp. are long rod-shaped heterotrophic bacteria, which belong to Bacteroidetes, and they mainly live in the bottom mud. They can degrade organics, such as organic acids and polysaccharides [44], and contribute to degradation in aerobic conditions, which make them suitable for bacterial mats and water depuration in bottom muds. They are also good in organic degradations in sludge, however, *Lewinella agarilytica* can degrade agar and xylan (polysaccharides) [45].

Figure 5 presents the composition of family levels of the bacterial biomes on the surfaces of each of the samples immersed during the winter season. The seven main bacterial flora observed in the summer season (Alteromonadaceae, Desulfobacteraceae, Desulfobulbaceae, Flavobacteriaceae, Helicobacteraceae, Piscirickettsiaceae, Rhodobacteraceae, and Saprospiraceae) were also observed in the winter season. These common biomes were present on the slag surfaces throughout the year. Helicobacteraceae was more abundant in slag II, which indicates that the slag has potential in seawater purification. In the winter season, almost all the bacteria are categorized in the unassigned genera of Helicobacteraceae, Desulfobulbaceae, and Saprospiraceae (Table 2).

This study elucidates the dominance of sulfide-oxidizing bacteria in seawater-immerged slag samples. In particular, Helicobacteraceae was found as the typically dominant biome even though its rate in the samples tested in the winter was less than that of the summer. Considering that biofouling (biofilm formation) in the summer season proceeded faster than that in the winter season, then Helicobacteraceae would grow dominantly on the steelmaking slag samples (slag I, II, and III). Helicobacteraceae is considered to play a significant role in the oxidization of sulfides and denitrification around the immerged slag environment. In the summer season, slag types II and III showed more abundant Helicobacteraceae and Desulfobulbaceae than slag I. Moreover, in the winter season, Helicobacteraceae was more abundant in slag II, suggesting that type II is most suitable for applications in seawater purification. 

Iijima et al. reported that some *Pseudoalteromonas* species (members of Pseudoalteromonadaceae family) isolated from biofilm were formed in the soil of the tideland [46]. In this study, Pseudoalteromonadaceae was not the main family, both in the summer season and winter season. Because we analyzed the biofilm (forming) bacteria on the steelmaking slags using NGS technology, there are two possibilities: the main biofilm-forming bacteria on the steelmaking slags would be quite different from those in the natural soil of tideland; the main biofilm-forming bacteria would be unculturable in the steelmaking slags.

In this study, we focused on the early stage of biofouling, i.e., the biofilm formation (within one week), on steelmaking slag in a marine environment. Based on the constituents of steelmaking slags, previous studies, and experimental results of the bacterial biomes from biofilms formed on steelmaking slags, we propose the mechanism of (relatively initial) biofilm formation as follows (Figure 6): (Step 1) because of the high basicity of the steelmaking slag, calcium-containing oxidative products (including free lime) are mainly eluted from the slag just after being immerged in seawater, resulting in an increase in the pH (becomes alkaline) on the surface of steelmaking slag; (Step 2) the formation of the conditioning films is then started, thus, organic compounds are derived from the seawater; (Step 3) several bacteria from seawater are trapped on the conditioning films; (Step 4) because steelmaking slags are a good supplier of iron ions, phosphorus compounds, manganese ions, and other minerals needed for bacterial living, the trapped bacteria grow, proliferate, and produce EPS, thus the formation of biofilms commences; (Step 5) as the inner environment of the formed biofilm is influenced by an increase in pH (mentioned above), alkaline-like bacteria grow dominantly there. Specific bacteria continue to grow and produce EPS, resulting in the formation of characteristic biofilms on the steelmaking slag.

## 3. Materials and Methods 

### 3.1. Specimens

Slag types I, II, and III were provided by Professor Ryo Inoue of Akita University, Japan. The compositions are summarized in Table 3. Slag types I and II were hot metal pretreatment slags, and slag type III was a converter slag. These slags were misshapen, and each sample was measured in three angles using a caliper and then, the volumes were estimated. In the summer season, the volumes of types I, II, and III were 6.7 ± 2.9, 8.5 ± 6.4, and 17.3 ± 7.8 cm^3^, respectively, and in the winter season, they were 5.4 ± 2.3, 2.3 ± 1.2, and 8.1 ± 3.3 cm^3^, respectively. Polyurethane sponge foams (9.0 cm^3^ in summer and 6.8 cm^3^ in winter) (Komeri, Niigata, Japan) were used as the reference specimens.

### 3.2. Marine Immersion Test

Each type of slag and reference specimen (10 units) was packed in a polyethylene mesh sack (mesh size: 4 mm, sack size: 15 × 30 cm). The packed sacks were tied to a steel sample holder with a polypropylene rope, and the holder was suspended on two thick polyethylene ropes (Figure 7a). This apparatus is referred to as an immersing holder. The immersing holder was attached to a floating dock at Marina Kawage in Suzuka, Mie, Japan (34°79.8327′ N, 136°56.2372′ E) (Figure 7b). The immersion tests were performed on 18–25 August 2017 (summer), and 22–29 December 2017 (winter). Some environmental parameters for the seawater are summarized in Table 4 as a reference. The data have been obtained from the reports produced by Mie Prefectural Fisheries Research [47,48]. The sample holder was maintained at a depth of 2 m below the water surface. On the third day of immersion, four units of each specimen were removed from the sacks. One was analyzed by Raman spectroscopy and others were stored at −80 °C until DNA extraction. On the seventh day, all specimens were removed. One of each type was analyzed by Raman spectroscopy and others were also stored at −80 °C.

### 3.3. Raman Spectroscopic Analysis of Biofilm

Microscopic Raman spectroscopic analysis of the slag samples after the marine immersion test for 3 or 7 days was conducted using a laser Raman spectrophotometer (NRS-3100, JASCO, Tokyo, Japan). The measurement points were selected using a microscope attached to the spectrophotometer, and these points were irradiated by the laser. The measurement range was 657–1688 cm^−1^. The measurements were performed for 10 s. Sixteen measurements were performed each, of which the average was recorded as the biofilm spectrum.

### 3.4. DNA Extraction

Each sample was pooled into a tube and homogenized using Beads Crusher (Taitec, Saitama, Japan) for 1 min at 4000 rpm. The debris was removed by quick centrifugation (1 s at 8000× *g*). The supernatant was collected and centrifuged for 20 min at 15,000× *g* to recover the microbial cells. The bacterial genomic DNA on each sample was extracted using a Promega DNA purification system (Promega, Madison, WI, USA) according to the manufacturer’s instructions.

### 3.5. Microbiome Analysis

The V1–V2 regions of 16S rRNA genes were amplified using Ex Taq Polymerase (Takara Biotechnology, Shiga, Japan). The first step of the polymerase chain reaction (PCR) was performed using the universal primers 27F (5′-ACACTCTTTCCCTACACGACGCTCTTCCGATCTAGRGTTTGATYMTGGCTCAG-3′) and 338R (5′-GTGACTGGAGTTCAGACGTGTGCTCTTCCGATCTTGCTGCCTCCCGTAGGAGT-3′). Thermal cycling was maintained for 24 cycles. The PCR products were purified using the Wizard SV Gel and the PCR Clean-Up System (Promega). The second step of the PCR was amplified using the purified products from the first PCR (12 cycles). A second PCR was performed, following the same method as those of the first PCR, using 27F and 338R primer-added barcodes for each sample. After the second PCR, the products were treated as described above for the first PCR products.

The samples were sequenced on an Illumina MiSeq platform. After using a FASTQ barcode splitter (FASTX toolkit) [49] to exclude any readings with base sequences other than the primer, the readings with a quality value of less than 20 or with a length of less than 40 base units were removed using sickle tools [50]. The sequences were grouped into operational taxonomic units at a 97% similarity level using the QIIME software package [51].

## 4. Conclusions

This study focused on the relatively initial formation of biofilms on steelmaking slags in a marine environment since it is a crucial step in biofouling in the establishment of seaweed beds. Three types of steelmaking slags, two hot metal pretreatment slags (slag I and slag II) and a converter slag (slag III), were immersed in a marine environment, and biofilms were successfully formed. Compared with the biofilms formed on the polyurethane sponge foams as a control specimen, those formed on the slag samples were rich in the S-oxidizing bacteria of Helicobacteraceae and Piscirickettsiaceae as well as in organic acid-oxidizing bacteria of Desulfobulbaceae. These bacteria can purify marine waters, therefore, the biofilms formed on the steelmaking slag could be suitable for marine water treatments. Furthermore, the biofilms on slag samples II and III were richer in Helicobacteraceae and Desulfobulbaceae than those on slag I, which indicates that slag samples II and III are superior to slag sample I in marine water treatments. However, the relationship between the components of the various slag types and the trends of the biofilm formation suitable for water treatment remains unknown. Thus, there is a need for more investigation on these in future research, which will focus on the components of steelmaking slags, particularly, the sulfur and calcium contents.

## Figures and Tables

**Figure 1 ijms-21-06945-f001:**
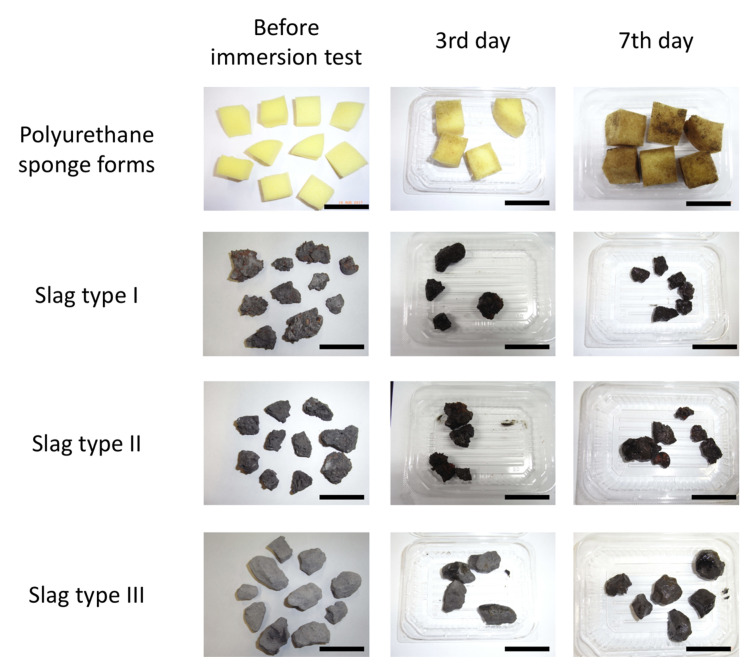
Appearance of samples before and after immersion during summer. Black bars indicate 5 cm.

**Figure 2 ijms-21-06945-f002:**
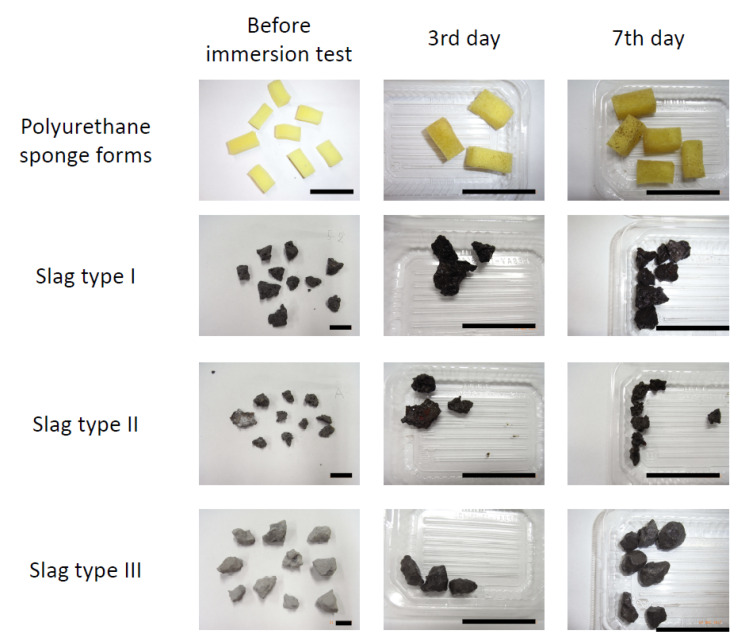
Appearance of samples before and after immersion during winter. Black bars indicate 5 cm.

**Figure 3 ijms-21-06945-f003:**
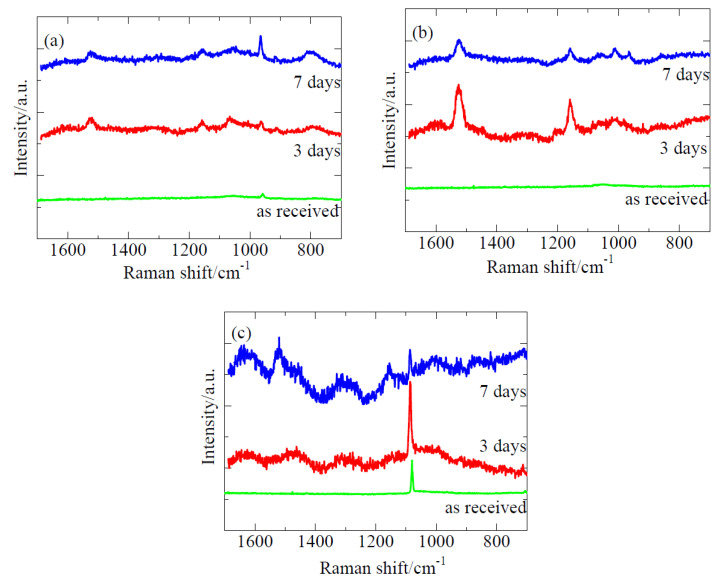
Raman spectra of (**a**) slag I, (**b**) slag II, and (**c**) slag III after the marine immersion test for 7 days (blue lines), 3 days (red lines), and as-received (green lines) in summer.

**Figure 4 ijms-21-06945-f004:**
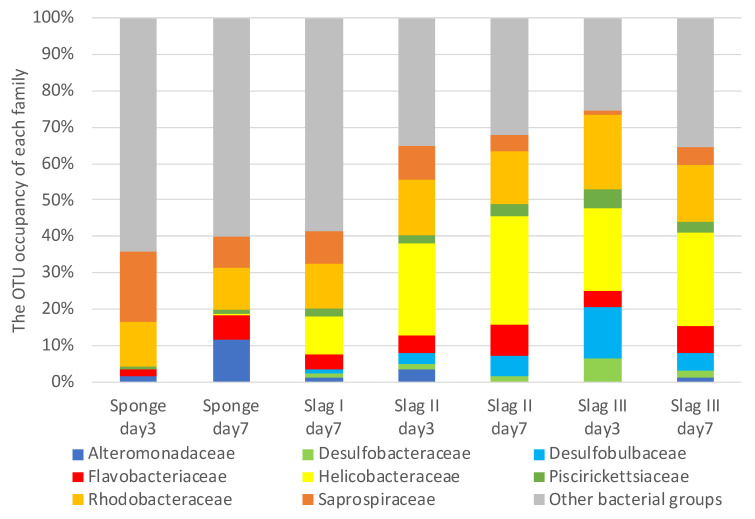
Bacterial biomes of three types of slag samples displayed at 3-day and 7-day incubation stages in the summer season. Artificial sponge was used as a control substrate.

**Figure 5 ijms-21-06945-f005:**
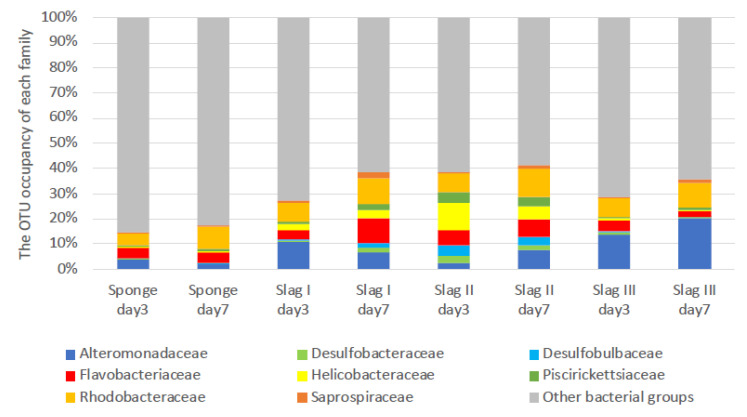
Bacterial biomes of three types of slag samples displayed at 3-day and 7-day incubation stages in the winter season. Artificial sponge was used as a control substrate.

**Figure 6 ijms-21-06945-f006:**
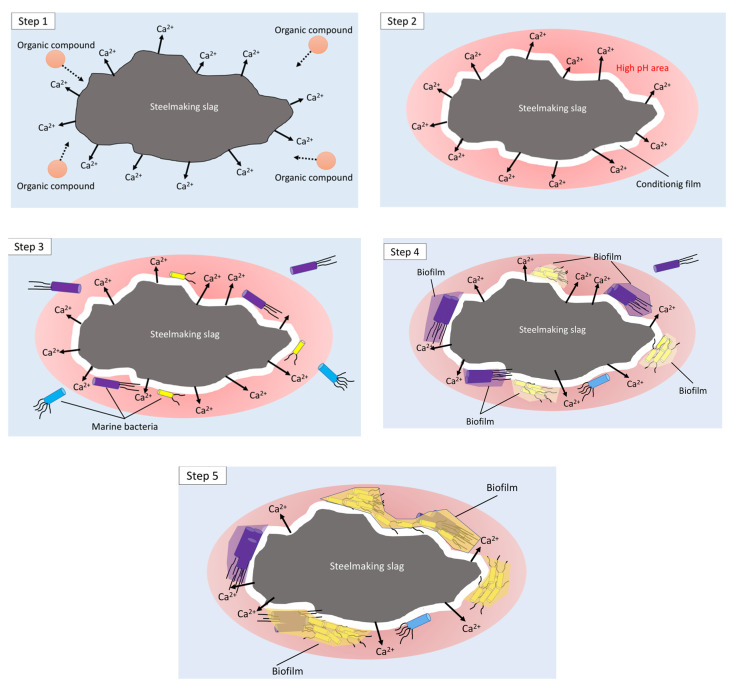
Proposed mechanism of biofilm formation on steelmaking slag in the marine environment.

**Figure 7 ijms-21-06945-f007:**
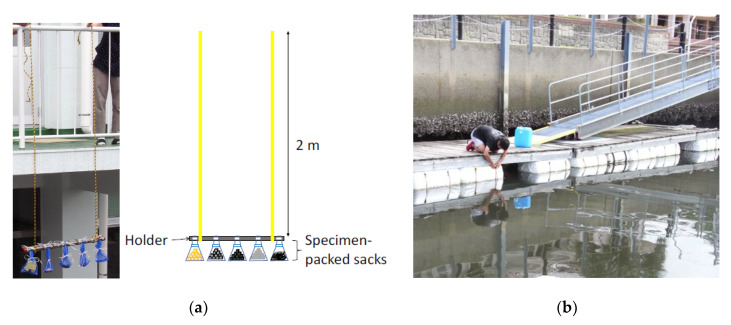
(**a**) The apparatus of the marine immersion test and (**b**) the location for the test.

**Table 1 ijms-21-06945-t001:** Percentage of the bacterial community in each sample among Helicobacteraceae, Desulfobulbaceae, and Saprospiraceae in the summer season. - = Not analyzed.

Family	Genus	Sponge Day 3 (%)	Sponge Day 7 (%)	Slag I Day 3 (%)	Slag I Day 7 (%)	Slag II Day 3 (%)	Slag II Day 7 (%)	Slag III Day 3 (%)	Slag III Day 7 (%)
Helicobacteraceae	*Sulfurimonas*	0	0	-	5	8	4	5	4
	unassigned	100	100	-	95	92	96	95	96
Desulfobulbaceae	unassigned	100	100	-	100	100	100	100	100
Saprospiraceae	*Lewinella*	92	78	-	93	92	94	89	91
	unassigned	8	22	-	7	8	6	11	9

**Table 2 ijms-21-06945-t002:** Percentage of the bacterial community of each sample in Helicobacteraceae, Desulfobulbaceae, and Saprospiraceae in the winter season.

Family	Genus	Sponge Day 3 (%)	Sponge Day 7 (%)	Slag I Day 3 (%)	Slag I Day 7 (%)	Slag II Day 3 (%)	Slag II Day 7 (%)	Slag III Day 3 (%)	Slag III Day 7 (%)
Helicobacteraceae	unassigned	100	100	100	100	100	100	100	100
Desulfobulbaceae	unassigned	100	100	100	100	100	100	100	100
Saprospiraceae	*Lewinella*	0	0	0	0	0	7	17	0
	unassigned	100	100	100	100	100	93	83	100

**Table 3 ijms-21-06945-t003:** Slag compositions. C/S indicates CaO/SiO_2_ (mass rate%) referred to as basicity.

Slag	Total CaO	f-CaO	SiO_2_	Al_2_O_3_	MgO	Total Fe	MnO	P_2_O_5_	S	C/S
I	37.6		22.6	4.2	6.5	4.7	12.0	5.4	0.02	1.67
II	55.3		18.7	3.0	1.9	4.3	5.6	4.6	0.29	2.91
III	52.2	15.9	14.1	3.0	2.8	12.2	4.3	2.9		3.71

**Table 4 ijms-21-06945-t004:** Environmental parameters for the seawater. The observation point was 34°49.88 N, 136°37.81 E.

Parameter	Summer ^(1)^	Winter ^(2)^
Temperature (°C)	27.98	14.83
Salinity (PSU) ^(3)^	26.167	20.903
COD (ppm)	3.49	0.38
NH_4_-N (μg-at./L)	0.60	0.42
NO_2_, _3_-N (μg-at./L)	1.15	3.23
DIN (μg-at./L)	1.76	3.65
PO_4_-P (μg-at./L)	0.07	0.34

^(1)^ The observation day was 10 August 2017; ^(2)^ the observation day was 4 December 2017. ^(3)^ PSU: Practical salinity unit. Temperature and salinity were recorded at a depth of 2 m, the others were recorded at sea surface (0 m).

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
