# Peer review of "Investigation of Biofilms Formed on Steelmaking Slags in Marine Environments for Water Depuration"

_ijms, 2020, doi:10.3390/ijms21186945_

Round 1

Reviewer 1 Report

The manuscript presented by Ogawa and co-authors describes the evaluation of the early stage of bacterial biofilm formation as well as its identification and population characterization, utilizing three types of steelmaking slag as substrate/matrix.

The use of alternative substrates for biofilm formation as well as the characterization of this process is a timely topic especially considering the putative use of bacteria in biotechnological processes.

Although the authors demonstrate the biofilm formation and the characterization of the bacterial flora present in these biofilms, some of the observations and, consequently, the conclusions presented, lack a more solid interpretation. In some points, additional experiments should be performed in order to validate some of the results obtained.

Therefore, the major points that need clarification by the authors are:

  • The rationale behind the two chosen timeframes (3 and 7 days) is not clear in the manuscript. As this is the first time these steelmaking slags are used in these conditions, surface analysis techniques such as AFM or SEM should be used to evaluate biofilm formation over time, from day 1 to day 15, for example. These experiments are essential to show the evolution of biofilm formation over time in the different slags which, in turn, is fundamental to decide the correct timeframes to collect the desired samples.

This is particularly important in the winter samples, where authors state that the microscopic Raman spectroscopic analysis was not sensitive enough to detect biofilm formation. In addition, the use of surface analysis techniques would also provide information regarding the topology of the biofilm and the bacterial coverage of the slags.

  • Although the use of Raman spectroscopy is well documented (and also in the manuscript references), the results obtained in this study are scarcely analyzed. As an example, page 4, line 108, “… these peaks (at around 1510 and 1150 cm–1) were derived from biofilm debris on the biofilms…”).

A detailed analysis of this section should be provided.

  • Another important point that should be better exploit is the type of bacteria that was found as well as their relative evolution in the two timeframes. Bacterial adsorption and biofilm formation in surfaces are fundamental to utilize a specific bacterium or a consortium of bacteria in biotechnological applications. In this study, a marine environment was used as the “culture media”. This aspect promoted the formation of biofilms containing bacteria consortium as demonstrated by the authors. Although the authors present putative roles of each genera based on previous studies, an integrated analysis of the genera found and the variation in their population in the two timeframes (3 and 7 days) would be helpful to correlate these with the three slags under testing and should be in the manuscript.

In this sense, the first time point of slag I is fundamental to the analysis of the experiment and cannot be “ignored”.

  • In addition, to better understand this issue, a comprehensive chemical analysis (hydrocarbons, metals, etc.) to the water that was used in the timeframe of the experiments should be presented in the manuscript.

In this sense, Figure 6 and the conclusion section should also be revised so that they contain better sustained statements.

Besides, some minor point should also be addressed:

  • 1, ln 33 – “… contains mainly lime (CaO)…” – authors should provide the correct chemical name when a chemical compound is mentioned. This should be revised and throughout the manuscript.

  • 4, ln 108 – The authors used reference 30 to explain a very important part of the Raman detection pattern observed and to correlate it with previous studies. However, ref. 30 is a paper in Japanese language that might be difficult to understand by the majority of the IJMS readers. A more suitable reference should be provided.

  • 5, ln 125 – “… The authors suggested that there was insufficient DNA obtained from the biofilms of slag I on day three…” – A comprehensive language editing should be performed throughout the manuscript.

Reviewer 2 Report

Reconstructions of the marine environment is an important issue, especially when artificial seaweed bed is needed. What is important, to grow seaweed the formation of the biofilm on an artificial seabed is crucial.

The Authors demonstrated that such a biofilm can be effectively created on a steelmaking slag.  They submerged three types of slag in the sea and analyzed the biofilm gained after 3 and 7 days.

The authors used Raman spectroscopy and PCR analysis to get information about the composition of the biofilm. The results are interesting and can be published after minor revision:

  • could the Authors demonstrate a SEM analysis of the surface of the slag, before and after? does the morphology of the slag influence the parameters of biofilm, e.g. thickness or its composition?
  • why 3 and 7 days? is is related to any particular value?
  • what is the mean size of the slag used in experiments? Figure 1 and Figure 2 lack the scale. Can the size or porosity influence the results?
  • what were the laser wavelength and power during the Raman measurements? and other details of measurements like optics and microscope? The authors should notice that Raman analysis would be challenging in this case, as the biofilm is a mixture of different bacteria. The use of SERS could help in terms of getting a better signal: to get that Authors need a SERS platform, in this case, a gold/silver nanoparticles should work. But it won't solve the problem of multiple signals from different bacteria.
  • the manuscript should be proof-read (e.g. a typo in line 77)

Round 2

Reviewer 1 Report

The revised version of the manuscript presented by Ogawa presents significant improvements when compared with the previous one.

I believe that most of the questions were answered either in the manuscript or in the response letter.

Thank you for your effort.

Author Response

Dear Prof. Reviewer,

We appreciate taking your precious time to review this manuscript. Owing to your effort, we could discuss more deeply each other and improve the manuscript as much as we could.

We hope that our report will give helpful information to the readers of IJMS.

Sincerely yours,

Akiko Ogawa